# Mathematical Models of Leaf Area Index and Yield for Grapevines Grown in the Turpan Area, Xinjiang, China

**Lijun Su** [1,*] , **Wanghai Tao** [1,*], **Yan Sun** [2], **Yuyang Shan** [2] **and Quanjiu Wang** [1,2]

[1] State Key Laboratory of Eco-hydraulics in Northwest Arid Region, Xi'an University of Technology, Xi'an 710048, China; wquanjiu@163.com
[2] Institute of Water Resources and Hydro-Electric Engineering, Xi'an University of Technology, Xi'an 710048, China; sunyan199058@126.com (Y.S.); syy031@126.com (Y.S.)
[*] Correspondence: sljun11@xaut.edu.cn (L.S.); xautsoilwater@163.com (W.T.)

**Abstract:** The Leaf Area Index (LAI) strongly influences crop biomass production and yields. The variation characteristic of LAI and the development of crop growth models can provide a theoretical basis for predicting crops' water consumption, fruit quality and yields. This paper analyzes the relationship between measurements of aboveground grape biomass and trends in LAI and dry biomass production in grapes grown in the Turpan area. The LAI changes in grapes were estimated using the modified logistic model, the modified Gaussian model, the log-normal model, the cubic polynomial model, and the Gaussian model. Universal models of LAI were established in which the applied irrigation quota was applied to calculate the maximum LAI. The relationship between the irrigation quota and biomass production, yields, and the harvest index was investigated. The developed models could accurately predict the LAI of grapevines grown in an extremely arid area. However, the Gaussian and cubic polynomial models produced less accurate results than the other models tested. The Michaelis–Menten model analyzed the relationship between biomass and LAI, providing a numerical method for predicting dynamic changes in grapevine LAI. Moreover, the crop biomass increased linearly with the irrigation quota for quotas between 6375 and 13,200 $m^3/hm$. This made it possible to describe the grape yield and harvest index with a quadratic polynomial function, which increases during the early stages of the growing season and then decreases. The analyses of the relationship between yield and harvest index provide important theoretical insights that can be used to improve water use efficiency in grape cultivation and to identify optimal irrigation quotas.

**Keywords:** simulation model; grape; LAI; biomass; yield; harvest index

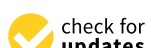



## 1. Introduction

Grapes are a valuable commercial crop used to produce both wine and raisins. Consequently, there is great interest in identifying the factors that govern their growth. It is known that grapevine yields and biomass growth are closely linked to their leaf area index (LAI, the total one-sided leaf area per unit of crop ground surface). The LAI influences biomass accumulation and transpiration, whilst the partitioning of the biomass affects the yield. Moreover, the LAI is a critical variable in the various process models, such as evapotranspiration and canopy photosynthesis. It also affects the size of the plant-atmosphere interface and thus plays a key role in the exchange of energy and mass between the canopy and the atmosphere. During the early stages of the growing season, LAI is low and increases slowly. However, as the season progresses, it begins to increase rapidly. It peaks before the leaves start to senesce, and the plants reach physiological maturity. Studies of trends in LAI can provide technical support for simulations of dynamic changes in grape biomass and yields. There have been several techniques to predict *LAI* using crop simulation models [1–3] and generic crop models [4–7].

A few input parameters should be necessary for an ideal simulation model for LAI development and crop yields. The model should be based on the underlying physiological

and phenological processes observed in real plants. Estimation [8–14] and species growth models [15,16] are two methods for this problem explored. Estimating methods rely on remote sensing [17–20] or direct measurements [10,21] to determine the LAI. On the other hand, species growth models are proposed by a theoretical foundation, such as the logistic model, the Gompertz model, the Richards model, and the Chanter model. One or more parameters in these models indicate some physical properties, and the models are used to describe how population sizes and biomass change over time. The logistic model is a well-known mathematical model for predicting population growth [22–26] with a high accuracy level. However, one of the major drawbacks of logistic models is that they can only depict growth under specified conditions because the values of their parameters change depending on the environment. Temperature, for example, affects the growth curve of bacteria. In principle, logistic models can be used to simulate any growth process. However, in this case, the relationship between the temperature and the bacterial growth rate means that such models cannot be summarized by creating a synthetic model [27]. When the temperature is included as a variable in the model, it can be used to predict population growth and restrictions imposed by environmental factors.

Growing degree days (GDD) is an important meteorological factor affecting the crop growth indexes such as plant height, leaf area index, biomass and harvest index [28]. To collect and analyze the crop growth data in 50 regions of China, Liu et al. [24], Wang et al. [29], Su et al. [30] and Liu et al. [31] proposed the normalized logistic models of potato, winter wheat, summer maize, rice, and cotton by using growing degree days as a key variable, respectively. Wang et al. [25] established the logistic cotton growth model under drip irrigation with film mulch in Xinjiang, China. They discussed the relationships between maximum leaf area index, maximum dry matter accumulation, the harvest index (HI), and total irrigation amount. Irrigation and nitrogen application are also the key environmental factors in crop growth. Overman et al. [32–36] used the logistic model to simulate the biomass production of forage grasses by incorporating harvest timing and water use efficiency in different years as functions of nutrition and water content. Yang et al. [37] used fruit growth data to develop an improved logistic model that explained the individual tomato fruits growth depending on the environmental parameters such as different planting densities and seasons. Munitz et al. [38] established the relationship between LAI and crop coefficient of mature *Vitis vinfera* cv., which was the basis for developing a comprehensive irrigation model considering climate conditions, canopy area and grapevine-specific characteristics. Chen et al. [39] used the logistic model to fit the dynamic change of summer maize leaf-area index based on GDD under different nitrogen, phosphorus and potassium nutrition levels.

The logistic model is usually used to simulate the crop growth available, but it is not appropriate for predicting the late stable and decline periods. A few mathematical models, such as the modified logistic model, the cubic polynomial model and the exponential growth model (also known as the Gaussian model, the modified Gaussian model and the log-normal model) [40,41], have been devised to address this flaw. However, these models are originally developed to simulate the growth process of maize [31,39,42,43], wheat [23,24,31,44] and cotton [25,45,46], and there have been few investigations into their applicability to Thompson Seedless grapevines. The leaves are the primary organs involved in photosynthesis and transpiration in grapevines, and dynamic changes in leaf mass have an impact on grape yields. Therefore research aimed at modelling dynamic changes in the aboveground biomass of grapevines can provide important data that has the potential to improve water use efficiency and predict grape yields.

The objective of this study was to develop an approach to simulate LAI and yield of grape plants using mathematical models and irrigation quotas. The LAI of grapevines in the Tu-ha basin was simulated using the modified logistic model, modified Gaussian model, log-normal model, Gaussian model and cubic polynomial model. The accuracy and practicality of the models were assessed in each situation. According to the Michaelis–Menten equation, a given crop's maximum dry matter yield can be estimated based on its

maximum LAI. Moreover, a relationship between grapes' harvest index and peak LAI was established. Finally, a mathematical model to predict grape yields in Turpan was developed based on the relationship between LAI and yield.

## 2. Materials and Methods

### 2.1. Experimental Fields

The experiments were conducted in an area used for table grape cultivation in TieTier village, located 12 km to the southeast of Turpan city (42.87° N, 89.20° E, 32.8 m above sea level) in Xinjiang, China. Extreme weather such as whole gale and low temperatures are experienced between February and June. The mean annual precipitation in this region is 16.5 mm, and the mean annual evaporation is 3600 mm. The experimental fields were planted with grapevines from the Thompson seedless cultivar. The vines were grown on trellises and were 12 years old at the start of the experiment. The cultivation pattern and the schematic drawing for grapevines are shown in Figures 1 and 2. The average spacing between adjacent rows was 1.2 m, and the average spacing between vines within a row was 3 m. The soil texture of the experimental field is clay loam and uniform within a one-meter depth. The average soil bulk density is 1.47 g/cm$^3$, and the average saturated soil water content is 0.39 cm$^3$/cm$^3$.

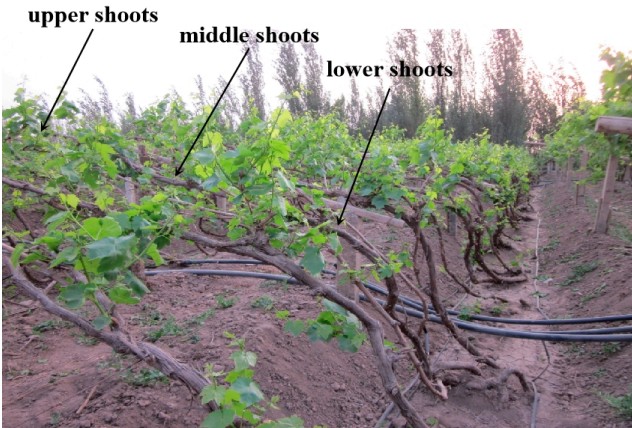

**Figure 1.** Cultivation pattern of the grapevines in the field. The principal growth stage is shoot development.

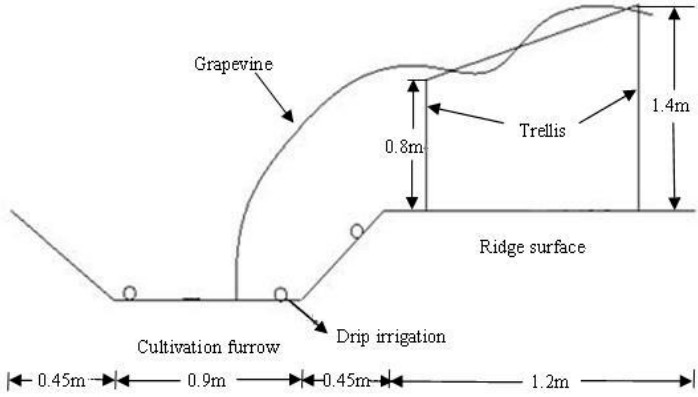

**Figure 2.** Schematic drawing of the grape planting structure in the field.

### 2.2. Experimental Design

To analyze the effect of different irrigation quotas on grape biomass and yields, six irrigation treatments based on surface drip irrigation (see Table 1) were evaluated. The irrigation scheme was set up with three water pipes in a single ditch with a drip flow of 2.7 L/h. Adjacent water droppers were separated by 40 cm. The first irrigation quota is

60 mm for every treatment before the growth period. The six treatments differed in terms of their irrigation frequency. Treatments X1 to X3 involved irrigation every 4.5, 9, and 13.5 days, respectively. In treatments X4 to X6, the irrigation frequency was higher during the period of high water demand than at other times. Thus, treatment involved irrigation every 4.5 days during the period of high water demand and every 9 days at other times; treatment X5 involved irrigation every 4.5 and 13 days, respectively; and treatment X6 involved irrigation every 9 and 13.5 days, respectively. Numerical data on the different irrigation schemes are presented in Table 1.

**Table 1.** The design of the irrigation experiments.

| Irrigation Treatment | Irrigation Frequency | | Irrigation Quota for Each Application (mm) | Irrigation Quota (mm) | Drip Irrigation Tape Parameters | |
| --- | --- | --- | --- | --- | --- | --- |
| | Critical Watering Time | Non-Critical Watering Time | | | Drip Flow (L/h) | Dripper Spacing (cm) |
| X1 | 4.5 | 4.5 | 52.5 | 1215 | | |
| X2 | 4.5 | 9 | 52.5 | 1005 | | |
| X3 | 4.5 | 13.5 | 52.5 | 900 | 2.7 | 40 |
| X4 | 9 | 9 | 52.5 | 847.5 | | |
| X5 | 9 | 13.5 | 52.5 | 690 | | |
| X6 | 13.5 | 13.5 | 52.5 | 637.5 | | |

New shoots start growing during the budding period and continue growing until the vine reaches maturity. The growth of all shoots on the experimental vines was recorded from their first appearance until the point when their growth stopped or became so slow that their length could be assumed to be constant. The results presented in this work are based on measurements acquired between 1 April and 1 October 2009. The growing season was divided into a period of critical water demand, which corresponds to the time the fruits inflate, and periods of non-critical water demand, which cover the remainder of the growing season.

Grape biomass data were collected from three grapevines in each treatment. The upper, lower, and middle shoots (see Figure 1) in the selected grapevines were chosen to record the growth characteristic indexes separately. The recorded indexes included: the number of shoots, the length of the shoot (measured by rulers), the number of leaves on each shoot, and the main vein length of recorded leaves (measured by rulers). Finally, nine groups of recorded data were averaged for each treatment. The method used to calculate LAI was defined by the function

$$\text{LAI} = \frac{\text{B} \times \text{L} \times \text{A}}{10.000 \times \text{S}} \tag{1}$$

where B is the number of shoots; L is the number of leaves; S is the average area covered by a single grapevine, m$^2$; and A is the average area of each leaf, cm$^2$ and can be estimated by the power function of the main vein length as shown in Figure 3.

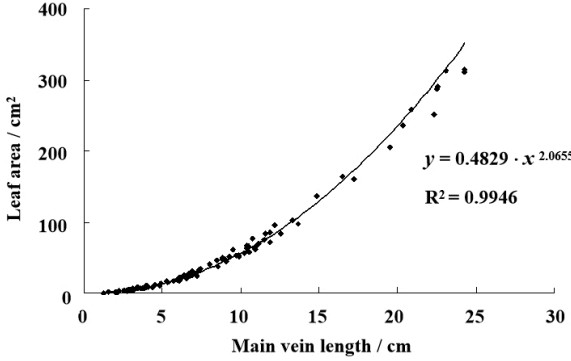

**Figure 3.** Relationship between the main vein length and the leaf area.

Aboveground dry matter mass was measured by the oven drying method. In each irrigation treatment, all shoots and leaves were collected in the unit areas, and the oven dried these samples. Because of the dried sample's mass in the unit areas, the aboveground dry matter mass can be estimated by multiplying by the treatment area.

### 2.3. GDD Calculations

The GDD was calculated by the difference between $T_{avg}$ and $T_{base}$ [24]:

$$GDD = \sum_i \left(T_{avg,i} - T_{base,i}\right) \tag{2}$$

where $T_{avg}$ is the daily average temperature, and $T_{base}$ is the base temperature that the crops need to grow. $T_{base} = 10\,^{\circ}C$ for the grapevines. $T_{avg}$ was calculated by the arithmetic mean of the daily maximum ($T_x$) and minimum ($T_n$) temperatures, subject to the limitation that it could not exceed $T_{upper}$ (the temperature above which no further increase in plant growth rate is observed, and $38\,^{\circ}C$ for the grapevines) or be less than $T_{base}$. The method proposed by the Food and Agriculture Organization (FAO) was used [47]:

$$T_{avg} = \frac{(T_x^* + T_n^*)}{2} \tag{3}$$

$$\begin{cases} T_x^* = T_{upper} & \text{if } T_x^* \geq T_{upper} \\ T_x^* = T_{base} & \text{if } T_x^* \leq T_{base} \\ T_x^* = T_x & \text{else} \end{cases} \tag{4}$$

$$T_n^* = T_{upper} \ \text{if } T_n^* \geq T_{upper} \tag{5}$$

### 2.4. Leaf Area Index Growth Models

In this paper, five mathematic models (the modified logistic model, the Gaussian model, the modified Gaussian model, the log-normal model and the cubic polynomial model) were used to describe the changes in the LAI over the experimental period. The modified logistic model [37,40,41] is defined as follows:

$$LAI = \frac{LAI_m}{1 + e^{a + b \cdot GDD + c \cdot GDD^2}} \tag{6}$$

where LAI is the leaf area index; $LAI_m$ is the maximum leaf area index; and GDD is growing degree days, $^{\circ}C$. a, b, and c are experience coefficients.

The log-normal model [41] is defined as follows:

$$LAI = LAI_m \exp\left[-0.5\left(\frac{\ln(GDD/GDD_0)}{\alpha}\right)^2\right] \tag{7}$$

where $LAI_m$ is the maximum leaf area index, $GDD_0$ represents the growing degree days on which the leaf area index reaches the maximum value, $^{\circ}C$. $\alpha$ is an experience coefficient. It can be seen from Equation (7) that when $GDD = GDD_0$, $LAI = LAI_m$.

The cubic polynomial model is defined as follows:

$$LAI = a_0 + a_1 GDD + a_2 GDD^2 + a_3 GDD^3 \tag{8}$$

where $a_0$, $a_1$, $a_2$, and $a_3$ are experience coefficients corresponding to meaningless empirical parameters.

The German mathematician Gauss proposed the Gaussian model, defined by Equation (9), and the model curve has a single peak and 3 parameters. The modified Gaussian model is derived from the Gaussian model and is defined by Equation (10) [41].

$$LAI = LAI_m \exp\left[-0.5\left(\frac{GDD - GDD_0}{\beta}\right)^2\right] \tag{9}$$

$$LAI = LAI_m \exp\left[-0.5\left(\frac{|GDD - GDD_0|}{\beta}\right)^\gamma\right] \tag{10}$$

where $\beta$ and $\gamma$ are experience coefficients. The parameters in the above models were obtained by the genetic algorithm as implemented in Matlab.

### 2.5. Statistical Analysis

The root-mean-square error, the coefficient of determination, and the relative error were used to evaluate the performance of the different models. The root-mean-square error (RMSE) is a statistical method used to analyze the deviation between measured and calculated values. The smaller the RMSE, the better the simulated results. The RMSE is defined by the function

$$RMSE = \sqrt{\frac{\sum_{i=1}^{n}(O_i - S_i)^2}{n}} \tag{11}$$

where $O_i$ is the measured values, $S_i$ is the predicted values, and n is the sample size.

The coefficient of determination ($R^2$) is another method for evaluating the difference between the measured and calculated values.

$$R^2 = 1 - \frac{\sum_{i=1}^{n}(O_i - S_i)^2}{\sum_{i=1}^{n}(O_i - \overline{O}_i)^2} \tag{12}$$

In this work, the relative error (Re) is defined as follows:

$$Re = \sqrt{\frac{\sum_{i=1}^{n}(O_i - S_i)^2}{\sum_{i=1}^{n}O_i^2}} \tag{13}$$

## 3. Results and Discussion

### 3.1. Leaf Area Index Simulation Models

LAI values calculated using Equation (1) for the different irrigation treatments are shown in Figure 4. While no treatments have identical LAI values in any growth period, the trends of all treatments in the LAI over growing degree days are similar: it increases rapidly between 200 °C and 1400 °C, then gradually between 1400 °C and 2200 °C. After this point, it starts to decrease slowly. As shown in Figure 4, there are two prunings in the grapevine's growth period. The first pruning begins when the GDD is over 400 °C; the second pruning begins when the GDD is over 700 °C. The maximum LAI was observed when GDD was about 1690 °C.

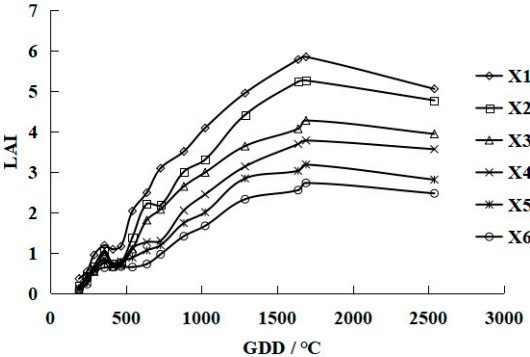

**Figure 4.** Relationship between leaf area index (LAI) and growing degree days (GDD) for different irrigation treatments.

We sought to identify a simple method for normalizing the LAI values to simplify subsequent analyses. This was done using the following equation:

$$RLAI = \frac{LAI}{LAI_m} \tag{14}$$

where RLAI denotes the relative LAI, and $LAI_m$ is the maximum LAI. Table 2 indicates that all treatments yielded different RLAI values in 2009. The highest standard deviation (0.0680) occurred on day 144 or growing degree day 883 °C, caused by pruning. Overall, the trends in RLAI were fairly similar for the six irrigation treatments. Consequently, a universal RLAI model was developed based on the mean RLAI values for all six treatments.

**Table 2.** Relative leaf area index (RLAI) values for the six different treatments at different points in time and mean RLAI values for all treatments in 2009.

| Time/Days | GDD/°C | RLAI | | | | | | Mean RLAI | Standard Deviation |
| | | X1 | X2 | X3 | X4 | X5 | X6 | | |
|---|---|---|---|---|---|---|---|---|---|
| 97 | 190 | 0.0619 | 0.0336 | 0.0250 | 0.0204 | 0.0416 | 0.0313 | 0.0356 | 0.0148 |
| 101 | 243 | 0.0927 | 0.0790 | 0.0954 | 0.1011 | 0.0835 | 0.0836 | 0.0892 | 0.0085 |
| 104 | 289 | 0.1619 | 0.1283 | 0.1263 | 0.1613 | 0.1734 | 0.2010 | 0.1587 | 0.0283 |
| 109 | 358 | 0.2025 | 0.1991 | 0.2328 | 0.2225 | 0.2397 | 0.2327 | 0.2215 | 0.0170 |
| 113 | 414 | 0.1860 | 0.1391 | 0.1551 | 0.1756 | 0.2258 | 0.2370 | 0.1864 | 0.0386 |
| 117 | 468 | 0.1988 | 0.1654 | 0.1659 | 0.1848 | 0.2467 | 0.2416 | 0.2006 | 0.0361 |
| 123 | 543 | 0.3480 | 0.2633 | 0.2561 | 0.2961 | 0.2776 | 0.2377 | 0.2798 | 0.0388 |
| 129 | 638 | 0.4253 | 0.4203 | 0.4237 | 0.3348 | 0.3331 | 0.2652 | 0.3671 | 0.0663 |
| 135 | 729 | 0.5289 | 0.4142 | 0.4857 | 0.4061 | 0.3732 | 0.3534 | 0.4269 | 0.0674 |
| 144 | 883 | 0.5094 | 0.4706 | 0.4946 | 0.4156 | 0.3377 | 0.3811 | 0.4348 | 0.0680 |
| 152 | 1024 | 0.6992 | 0.6283 | 0.6994 | 0.6122 | 0.6297 | 0.6112 | 0.6467 | 0.0415 |
| 165 | 1288 | 0.8467 | 0.8383 | 0.8518 | 0.8294 | 0.8921 | 0.8525 | 0.8518 | 0.0216 |
| 182 | 1639 | 0.9885 | 0.9943 | 0.9519 | 0.9766 | 0.9513 | 0.9351 | 0.9663 | 0.0236 |
| 185 | 1691 | 1.0000 | 1.0000 | 1.0000 | 1.0000 | 1.0000 | 1.0000 | 1.0000 | 0 |
| 217 | 2540 | 0.8644 | 0.9067 | 0.9223 | 0.9427 | 0.8826 | 0.9053 | 0.9040 | 0.0278 |

The mean RLAI values were fitted using the leaf area index growth models. The genetic algorithm was used to fit the model's parameters, and the fitted results of five models were shown in Figure 5. It can be seen from Figure 5 that the results of the models agreed well with the observed data of LAI, especially during the late stable period and the period of decline; in all cases, the correlation of determination between the fitted data and the experimental data were above 0.96.

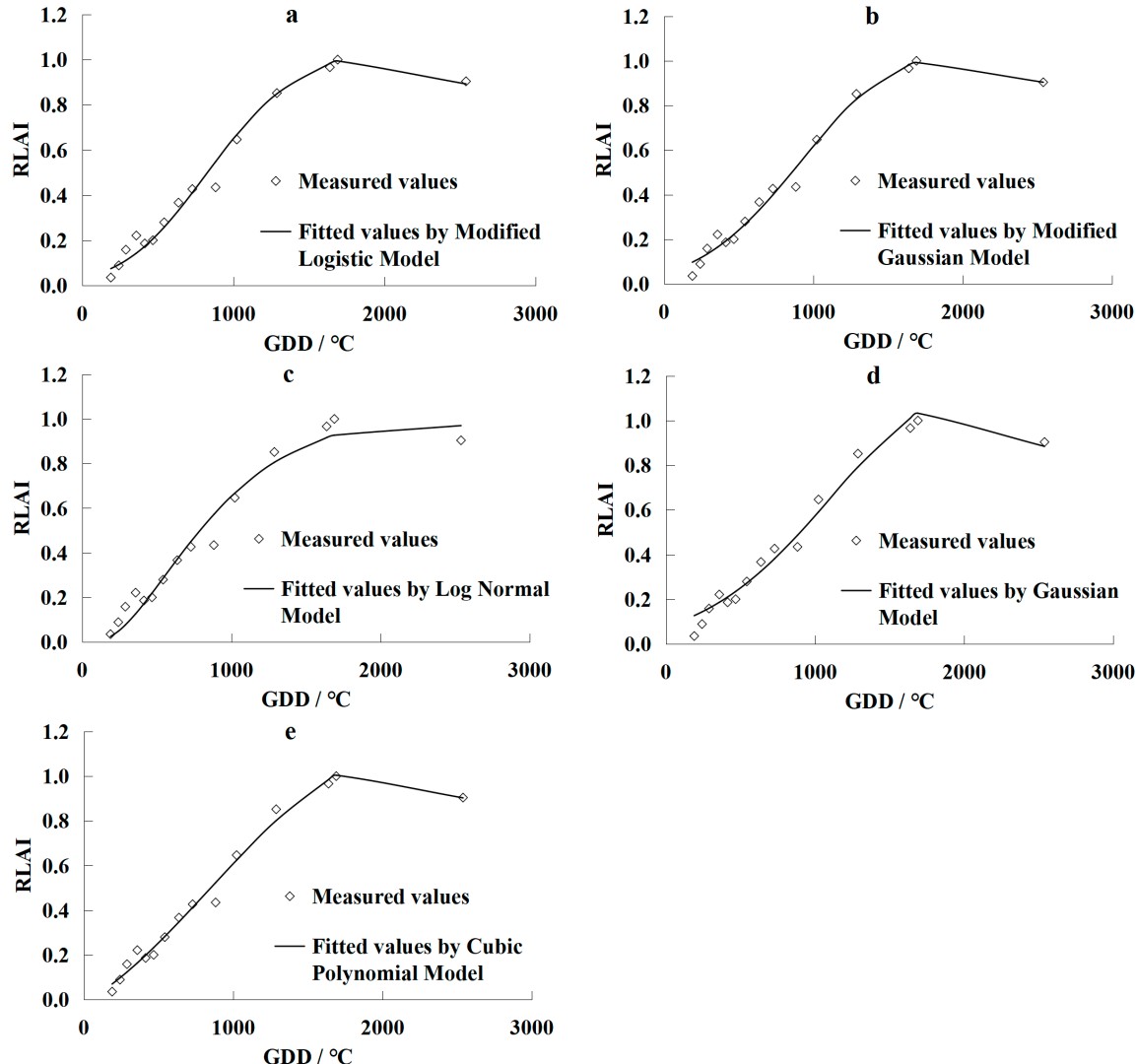

**Figure 5.** Comparisons of the measured and simulated mean relative leaf area index (RLAI): (**a**) modified Logistic model; (**b**) modified Gaussian model; (**c**) log-normal model; (**d**) Gaussian model; (**e**) cubic polynomial model. Measured values are the average RLAI values of six treatments, and simulated values are calculated by the five models and growing degree days (GDD).

Table 3 shows the fitted result of experience parameters in the five models. The coefficient of determination of all models is larger than 0.967, and the relative errors are smaller than 10.6%. This indicates that the simulated results of models are in good agreement with the measured data in the late stable period and the declining period of growth. Overall, the modified Gaussian model appears to be more accurate than the other models.

**Table 3.** Fitted values and simulated errors of five leaf area index growth models. Re is the relative error, R2 is the coefficient of determination, and RMSE is the root-mean-square error.

| Model | Expression | Re/% | $R^2$ | RMSE | Parameter Number |
|---|---|---|---|---|---|
| Modified Logistic Model | $\text{RLAI} = \dfrac{1.4}{1+e^{3.7437-4.8303\times10^{-3}\cdot\text{GDD}+1.2340\times10^{-6}\cdot\text{GDD}^2}}$ | 7.45 | 0.9837 | 0.0414 | 4 |
| Modified Gaussian Model | $\text{RLAI} = 1.0146 \cdot \exp\left[-0.5\left(\dfrac{|\text{GDD}-1983.1|}{983.96}\right)^{2.5757}\right]$ | 6.49 | 0.9877 | 0.0361 | 4 |
| Log-Normal Model | $\text{RLAI} = 0.9771 \cdot \exp\left[-0.5\left(\dfrac{\ln(\text{GDD}/2274.5)}{0.9142}\right)^2\right]$ | 8.16 | 0.9805 | 0.0454 | 3 |
| Cubic Polynomial Model | $\text{RLAI} = -1.6672\times10^{-10}\cdot\text{GDD}^3 + 4.2137\times10^{-7}\cdot\text{GDD}^2$ $+3.6610\times10^{-4}\cdot\text{GDD} - 0.01356$ | 6.37 | 0.9881 | 0.0354 | 4 |
| Gaussian Model | $\text{RLAI} = 1.0924 \cdot \exp\left[-0.5\left(\dfrac{\text{GDD}-1979.6}{862.44}\right)^2\right]$ | 10.60 | 0.9671 | 0.0589 | 3 |

The modified Gaussian, log-normal and Gaussian models are the exponential functions. The parameter $\text{GDD}_0$ in models is the growing degree days at the maximum relative leaf area index ($\text{RLAI}_{max} = 1$). The deviation between the value of $\text{LAI}_{max}$ in the three models is 0.0146 for the modified Gaussian model, $-0.0229$ for the log-normal model, and 0.0924 for the Gaussian model. That is to say, the predicted $\text{LAI}_{max}$ value calculated by the modified Gaussian model is smaller than the measured value. In contrast, those obtained with the log-normal and Gaussian models are larger. However, it should be noted that the absolute deviation of $\text{LAI}_{max}$ can represent the predictive accuracy, and the modified Gaussian model has the best performance in this respect.

The parameter number is highly sensitive to the flexibility and applicability of a model. The fewer parameters have the potential to increase the applicability of the model but may decrease the accuracy of its predicting results. As shown in Table 3, there are three parameters in the log-normal and Gaussian models and four parameters in the modified logistic model, the modified Gaussian model and the cubic polynomial model. However, the greater number of parameters makes them more complex to obtain. Therefore, in cases where high precision is not required, it may be preferable to use the log-normal model to simulate LAI.

### 3.2. The Relationship between Water Consumption and LAI

Equations (6) and (9) indicate that the maximum leaf area index, $\text{LAI}_m$, is an important parameter when simulating changes in LAI. Therefore, the general applicability of the models presented is dependent on the ability to quickly and easily calculate $\text{LAI}_m$. However, the leaf area index is sensitive to a number of factors, including the temperature, the number of growing degree days that have elapsed [41], the applied irrigation regime, water consumption, and so on. Some of these factors are difficult to measure regularly, so the ability to compute the LAI using readily obtained data plays a very important role in making a given method useful in day-to-day work. For example, $\text{LAI}_m$ can be directly measured when the GDD is about 1690 °C, as shown in Table 2. We selected the water consumption to estimate the $\text{LAI}_m$. The relationship between the water consumption and $\text{LAI}_m$, as determined by analysis of the results obtained in this work is presented in Figure 6.

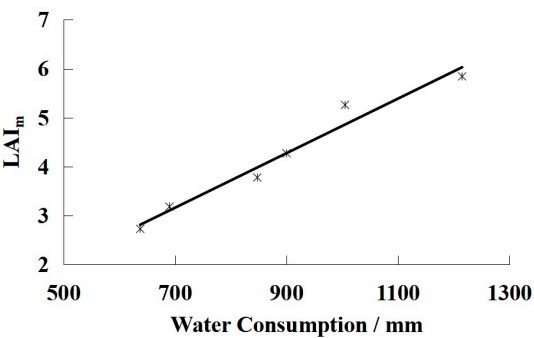

**Figure 6.** The relationship between leaf area index (LAI_m) and water consumption.

It is clear that the six tested irrigation regimes all yielded different $LAI_m$ values and that $LAI_m$ increases in direct proportion to the volume of water applied. The relationship between $LAI_m$ and the irrigation regime is described by the following linear equation:

$$LAI_m = 0.0056 \cdot WS - 0.7418 \tag{15}$$

where $LAI_m$ is the theoretical maximum LAI value, and WS is the water consumption. The coefficient of determination of the relationship between the measured and fitted values is 0.9654.

If we let

$$LAI_m = 0 \tag{16}$$

it follows that

$$WS = 132.46 \tag{17}$$

This means that if $WS \leq 132.46$, $LAI_m = 0$. Therefore, to achieve a non-zero LAI during the growth period, the volume of water consumption must exceed 132.46 mm.

The relationship between LAI and the water consumption can be determined using Equations (14) and (15):

$$LAI = RLAI \times LAI_m = RLAI \times (0.0056 \cdot WS - 0.7418) \tag{18}$$

where RLAI is the relative LAI, and WS is the water consumption. By combining the models in Table 2 with Equation (18), the universal mathematical models of LAI can be established. For example, the modified logistic model for calculating the LAI would be as follows:

$$LAI = \frac{1.4}{1 + e^{3.7437 - 4.8303 \times 10^{-3} \cdot GDD + 1.2340 \times 10^{-6} \cdot GDD^2}} \cdot (0.0056 \cdot WS - 0.7418) \tag{19}$$

Equation (19) indicates that the LAI of grapevines over the growing season can be estimated based on the applied irrigation regime and GDD. Figure 7 shows the fitted results for LAI in Turpan calculated by five developed universal models. The fitted results for the LAI values of grapevines grown in the Turpan area under the different irrigation regimes tested are shown in Table 3.

Figure 7 shows a good agreement between the fitted and measured grape leaf area index. The $R^2$ values of the fitted LAI values for five developed universal models are 0.97, 0.98, 0.96, 0.97 and 0.98, respectively. The relative errors for these five models are 10.02%, 9.75%, 12.11%, 11.28% and 9.49%, respectively. The RMSE for these five models are 0.9185, 0.8932, 1.1097, 1.0337 and 0.8696, respectively. Thus, the modified logistic or modified Gaussian model should be considered when fitting the LAI values of grapevines in the Turpan area.

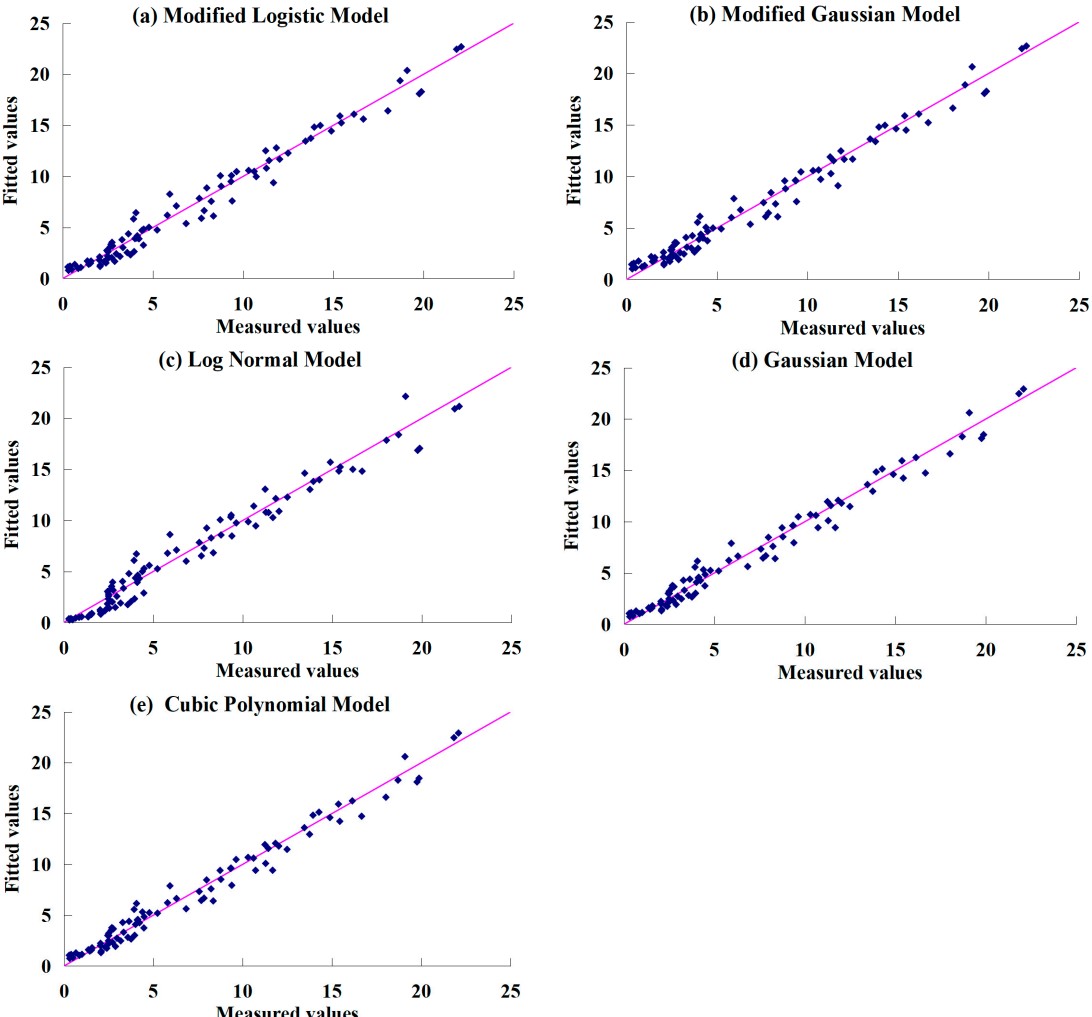

**Figure 7.** Comparisons of the measured and fitted leaf area index values are calculated by Equation (11) and models in Table 2: (**a**) modified Logistic model; (**b**) modified Gaussian model; (**c**) log-normal model; (**d**) Gaussian model; (**e**) cubic polynomial model.

### 3.3. Relationship between LAI and Dry Mass

Because the LAI is close to the photosynthesis and transpiration of plants, the rate of dry matter production and aboveground dry biomass of the plants is affected by the LAI. According to the differences in ecological and environmental factors, the rate of dry mass increases will be affected by the study location and growth period. However, the increases in dry matter mass will lead to increased LAI values in any growth period. Figure 8 shows the relationship between $LAI_m$ and measured relative dry biomass of treatments X1–X6 in 2009.

Figure 8 indicates a correlation between LAI and aboveground dry biomass, which can be described using the Michaelis–Menten equation. This means that the rate of increase in LAI decreases as the amount of dry mass increases. The measured data of dry matter and LAI in different growth periods are used to build the mathematical model.

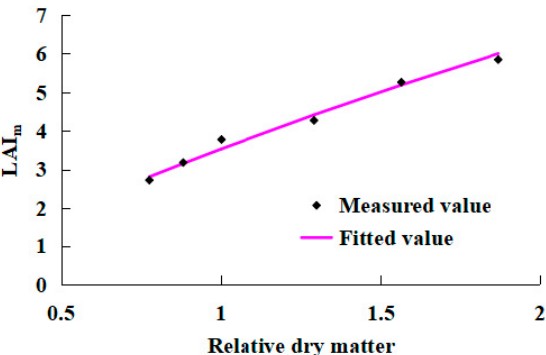

**Figure 8.** Relationship between maximum leaf area index (LAI$_m$) and relative dry biomass for treatments X1–X6 in 2009.

$$RM(t) = M(t)/M_0 \tag{20}$$

$$LAI(t) = \frac{P \cdot RM(t)}{1 + Q \cdot RM(t)} \tag{21}$$

where RM is the relative dry biomass; M is the aboveground dry biomass; $M_0$ is the aboveground dry biomass in a given growth period (here, $M_0$ = 29.33 t/ha, which is the measured dry mass for X4); t is the time relative to the start of the growing period, in days; and P and Q are experience parameters. To fit the values of P and Q, Equation (21) is converted to a linear form as follows:

$$\frac{1}{LAI(t)} = \frac{1}{P \cdot RM(t)} + \frac{Q}{P} \tag{22}$$

The least-squares method is used to analyze the linear relationship of $LAI^{-1}$ and $RM^{-1}$ in Equation (22), and the values of P and Q can be calculated by the curve fitting method. Thus, the mathematical model in Figure 7 is defined as the following equation:

$$LAI = \frac{3.9981 \cdot RM}{1 + 0.1314 \cdot RM} \tag{23}$$

where the sample number is 6, and the coefficient of determination is 0.97. The difference between the measured and fitted values is not significant ($p > 0.05$).

### 3.4. Mathematical Model of Yields

The data on crop water productivity in the simulations indicate that the aboveground biomass is determined by the amount of transpiration that occurs. The crop water productivity is a measure of the aboveground dry matter (g or kg) produced per unit of land area (m$^2$ or ha) per unit of water transpired (mm). The relationship between biomass production and water consumption is highly linear [48,49]. Figure 9 shows the relationship between biomass and water consumption.

As shown in Figure 9, the total biomass increases linearly with the irrigation quota according to the following equation:

$$B = 0.0683 \cdot I + 36.476 \qquad R^2 = 0.9534 \tag{24}$$

The LAI is related to the amount and rate of transpiration, which is related to biomass production by the water productivity parameter. The harvest index (HI) is defined to describe the harvestable portion of this biomass (yield):

$$Y = B \cdot HI \tag{25}$$

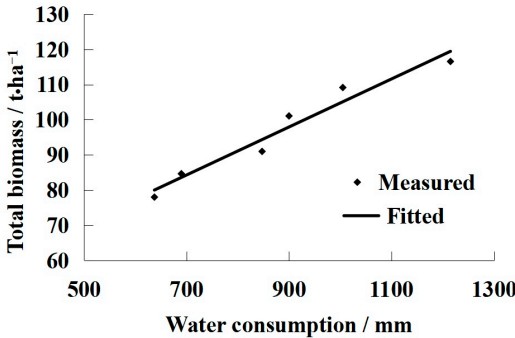

**Figure 9.** Relationship between total biomass (dry mass and yield) and water consumption.

When the total biomass (B) has been measured, the yield will be calculated by B and the harvest index Equation (18).

In this paper, the total biomass is divided into dry matter (grapevine) and yield (fruit).

$$B = M_m + Y \tag{26}$$

where B is the total biomass, $M_m$ is the maximum value of dry mass, and Y is the yield. Based on Equations (25) and (26), the yield can be calculated by the following equation:

$$Y = \frac{HI}{1 - HI} \cdot M_m \tag{27}$$

where HI is the harvest index.

According to Equation (23), the relationship between $M_m$ and $LAI_m$ is defined as follows:

$$M_m = \frac{LAI_m}{P - Q \cdot LAI_m} \cdot M_0 \tag{28}$$

The mathematical model of grape yields is established based on the parameter $LAI_m$. The relationship between HI and $LAI_m$ is shown in Figure 10 and is defined by Equation (29):

$$HI = -0.0071 \cdot LAI_m{}^2 + 0.0033 \cdot LAI_m + 0.7556 \qquad R^2 = 0.99 \tag{29}$$

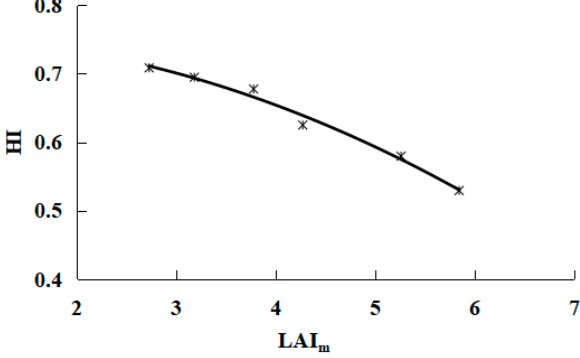

**Figure 10.** The relationship between the harvest index and maximum leaf area index ($LAI_m$).

Based on Equations (27)–(29), the mathematical yield model was defined as follows:

$$Y = \frac{-0.0071 \cdot LAI_m^2 + 0.0033 \cdot LAI_m + 0.7556}{0.0071 \cdot LAI_m^2 - 0.0033 \cdot LAI_m + 0.2444} \cdot \frac{LAI_m}{3.9981 - 0.1314 \cdot LAI_m} \tag{30}$$

Using the models described in Table 2 and the GDD at the maximum value of LAI, the $LAI_m$ can be fitted and used as input for the yield model. The fitted results of $LAI_m$ obtained by doing this are shown in Table 4. The statistical analysis results showed that the cubic polynomial model had the best precision of the other models. However, the cubic polynomial model parameters are meaningless, making the model lack application value. Thus, it is readily apparent that the modified logistic and modified Gaussian models give more accurate results than the alternatives.

**Table 4.** Predicted maximum leaf area index ($LAI_m$) values and associated statistical parameters for the different models considered in this work. Re is the relative error, $R^2$ is the coefficient of determination, and RMSE is the root-mean-square error.

| | | Predicted $LAI_m$ Value | | | | |
|---|---|---|---|---|---|---|
| | **Measured $LAI_m$ Value** | **Modified Logistic Model** | **Modified Gaussian Model** | **Log Normal Model** | **Cubic Polynomial Model** | **Gaussian Model** |
| X1 | 5.84 | 5.81 | 5.67 | 5.42 | 5.87 | 6.04 |
| X2 | 5.26 | 5.23 | 5.11 | 4.88 | 5.28 | 5.43 |
| X3 | 4.27 | 4.25 | 4.15 | 3.96 | 4.29 | 4.41 |
| X4 | 3.78 | 3.75 | 3.67 | 3.50 | 3.79 | 3.90 |
| X5 | 3.18 | 3.16 | 3.09 | 2.95 | 3.19 | 3.29 |
| X6 | 2.73 | 2.71 | 2.65 | 2.53 | 2.74 | 2.82 |
| Re/% | | 0.59 | 2.91 | 7.29 | 0.43 | 3.29 |
| RMSE | | 0.0255 | 0.1258 | 0.3149 | 0.0184 | 0.1421 |
| $R^2$ | | 0.9995 | 0.9868 | 0.9175 | 0.9997 | 0.9832 |

The grape yields achieved using the tested water treatments in Turpan can be fitted using the models described in Table 2 and Equation (30), giving the results shown in Table 5. The relative errors of all models are lower than 3.09%, and it is an adequate precision for predicting the yields. However, the output of the modified logistic model and modified Gaussian model correlates better with the experimental data than that of the log-normal and Gaussian models. It, therefore, seems that the modified logistic model or modified Gaussian model is optimal for fitting the peak leaf area index and grape yield in this case.

**Table 5.** Predicted grape yields and associated statistical parameters for the different models when used in conjunction with Equation (30). Re is the relative error.

| | | Predicted Yield Value | | | | |
|---|---|---|---|---|---|---|
| | **Measured Yield Value** | **Modified Logistic Model** | **Modified Gaussian Model** | **Log Normal Model** | **Cubic Polynomial Model** | **Gaussian Model** |
| X1 | 61.7 | 60.65 | 61.48 | 62.83 | 60.26 | 59.09 |
| X2 | 63.3 | 63.64 | 64.07 | 64.67 | 63.43 | 62.76 |
| X3 | 63.2 | 64.81 | 64.61 | 64.06 | 64.87 | 65.00 |
| X4 | 61.7 | 63.19 | 62.73 | 61.71 | 63.38 | 63.84 |
| X5 | 58.8 | 59.00 | 58.30 | 56.86 | 59.30 | 60.09 |
| X6 | 55.3 | 54.06 | 53.25 | 51.63 | 54.41 | 55.36 |
| Re/% | | 1.85 | 1.92 | 3.09 | 1.99 | 2.74 |

The results in Table 5 show that the mathematical yield models, which rely on only a single parameter, $LAI_m$, can be applied to estimate grape yields in Turpan with an acceptable level of accuracy. While any of the five models described in Table 2 can be used to obtain a reasonably accurate estimate of LAI, the most reliable estimated grape yields and fitted trends in LAI are achieved with the modified logistic model or modified Gaussian model.

## 4. Conclusions

Our analyses of aboveground grape LAI and yield data for different irrigation treatments in the Turpan area revealed that:

(1) Normalizing the measured LAI values makes it possible to disregard the impacts of irrigation quotas on the changes in grapevine LAI. The Linthe universal models were developed by the modified logistic model, the modified Gaussian model, the log-normal model, the Gaussian model, and the cubic polynomial model. Results using these models showed that they accurately fitted the measured data of LAI over the grapevines growing season in Turpan. However, the Gaussian and log-normal models yielded less accurate results than the other three models;

(2) Universal LAI models were developed to describe the relationship between the peak LAI value and water consumption. The models can be used to fit the dynamic changes of LAI over the growing season for different drip irrigation regimes. To ensure the yields of grapevine during the growth period, the water consumption must be at least 132.46 mm in the Turpan area;

(3) When the water consumption was in the range of 637.5 mm—11,215 mm, the biomass increased linearly, and the harvest index for the grapes was a quadratic polynomial function of the peak leaf area index. According to the relationships between yield, dry matter and harvest index, a mathematical yield model was proposed that relies on a single parameter: the peak leaf area index. Such descriptions of the relationship between yields and the harvest index can provide important information on improving water use efficiency.

**Author Contributions:** Conceptualization, L.S.; Funding acquisition, Y.S. (Yan Sun); Investigation, W.T.; Methodology, L.S., Y.S. (Yuyang Shan) and Q.W.; Supervision, Q.W.; Validation, Y.S. (Yan Sun) and Y.S. (Yuyang Shan); Writing—original draft, L.S.; Writing—review & editing, W.T. All authors have read and agreed to the published version of the manuscript.

**Funding:** This work was supported by the National Natural Science Foundation of China (51979220, 52109064, 41907010, 52179042), the Major Science and Technology Projects of the XPCC (2021AA003-2), the Fundamental Research Funds for the Central Universities (300102122105), and Natural Science Basic Research Plan in Shaanxi Province of China (No.2021JM-320).

**Data Availability Statement:** Not applicable.

**Acknowledgments:** The authors would like to thank Xinjiang institute of water resources and hydropower research for supporting the experimental conditions of this research. We also would like to thank Xing Wang in Xi'an university of Technology and Li Zhang in Changan University for supporting the simulation program of this research.

**Conflicts of Interest:** The authors declare no conflict of interest.

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
