# Peer review of "Mathematical Models of Leaf Area Index and Yield for Grapevines Grown in the Turpan Area, Xinjiang, China"

_agronomy, doi:10.3390/agronomy12050988_

Round 1
Reviewer 1 Report
GENERAL COMMENT
First of all, I would like to highlight the amount of work done by the authors. I consider this work a usable proposal with a medium to high impact on the scientific community.
However, there are two general points that I think should be improved. First of all, the references you use both in the introduction and in the discussion are too general. There is a lot of literature regarding the LAI and other parameters that are used in the specific case of the vineyard, and I believe that the authors must make an effort to search for and reinforce both their starting point (introduction) and the discussion with more specific references. This will considerably improve the manuscript.
On the other hand, the LAI estimation method used in Eq.1 must be justified, since the values ​​are used throughout the manuscript. In my opinion, LAI values ​​greater than 5 in vineyards are very rare. This must be justified, or, if there was an error, recalculate all models in the manuscript.
For these two reasons, I believe that a thorough review by the authors should be carried out.
There are some typos that I haven't corrected, but specific comments can be found below.
SPECIFIC COMMENTS
Introduction
Please define the leaf area index when it’s first cited. It’s usually defined as one-sided leaf area per unit of crop ground surface (m2m-2).
Please look for specific references to vineyards.
Materials and Methods
2.1 Experimental site
The sentence “The mean annual precipitation in this region is 16.5mm and the mean annual evaporation is 3600mm” is repeated in the introduction section, please remove one of them.
Please add the space between vines within a row.
Figure 1. Add growth stage according to BBCH scale at the moment of the picture or at least the date.
2.2 Experimental design
“In each case, the masses of the grapevine on the upper, lower, and middle shoots were recorded separately.” What do you mean with upper, lower, and middle shoots? Do you differentiate them for LAI estimation? If the calculation has been made with the average of the three specific locations, it should be made clear in the text. The trellis system used is not a conventional one, so a schematic of shoot locations can be useful for readers.
“The method used to calculate LAI was defined by the function (1)”: Where did this method come from? Please add references. Also, you have measured “the main vein length of recorded leaves”. How do you transform this parameter to the leaf area used in Eq. 1?
2.4 Leaf area index growth models
In the description of Equation 6, it says “k is the correction coefficient”, but there is no k in the formula.
In the equations, they name the experience coefficients as a, b and c. Sometimes different a, sometimes just b. Please be consistent in the way you express the equations. If they are not the same coefficient you can use different Greek letters, for example.
Results and discussion
3.1 Leaf area index simulation model
Here comes my major concern: Has the LAI estimation method (Eq 1) been validated in the study area? According to my experience and the bibliography consulted, the LAI on vineyards is up to 5 in very high dense vineyards. Thus, your results (LAIs up to 20) in an extremely arid region don’t seem realistic. Please review the formula used, or justified these results. You might need to consult more bibliography to justify your results. This applies to all the results from here on then measured LAI is used.
In Figure 3, please add to the footnote what LAI and CGDD/C mean. Figures and tables should be understandable without looking at the text. This comment applies to all Figures.
In Table 3: I suppose Re/% is the relative or simulated error, please clarify in the table text. Maybe it can be used: E (%). (Also in tables 4 and 5).
3.2. The relationship between water consumption and LAI
In this section, it is stated: “Some of these factors are difficult to measure regularly, so the ability to compute the LAI using data that is readily obtained plays a very important role in making a given method useful in day-to-day work”. Can you provide methods, tools, or recommendations, at least based on the bibliography, to estimate the maximum LAI in a practical way?
3.3. Relationship between LAI and dry mass
In Figure 7 footnote says: “Relationship between LAI and relative dry biomass for treatment X2 in 2009”. I understand this is for treatments X1-X6.
3.4. Mathematical model of yields
At the end of this section, it is said: “While any of the five models described in Table 2 can be used to obtain a reasonably accurate estimate of LAIm”. Table 2 is referred to the relative leaf area index (RLAI) and no model is described. Moreover, LAIm is the highest value of calculated LAI through Eq. 1, thus it is not estimated by the models presented.
Conclusions
Highlight the minimum water consumption for the studied area.
Author Response
Response to Reviewer 2 Comments
Point 1: First of all, the references you use both in the introduction and in the discussion are too general. There is a lot of literature regarding the LAI and other parameters that are used in the specific case of the vineyard, and I believe that the authors must make an effort to search for and reinforce both their starting point (introduction) and the discussion with more specific references. This will considerably improve the manuscript.
Response 1: Thank you for your suggestion. I have revised the introduction and replaced the older references by the new references in recent ten years.
Point 2: On the other hand, the LAI estimation method used in Eq.1 must be justified, since the values are used throughout the manuscript. In my opinion, LAI values greater than 5 in vineyards are very rare. This must be justified, or, if there was an error, recalculate all models in the manuscript.
Response 2: Thank you for your suggestion. Eq. (1) was wrong and LAI should be the value of total leaf area divided by covered area. I checked the data and recalculated the LAI by the following equation.
Point 3: Please define the leaf area index when it’s first cited. It’s usually defined as one-sided leaf area per unit of crop ground surface (m2m-2). Please look for specific references to vineyards.
Response 3: Thank you for your suggestion. I defined the LAI in the first paragraph in Introduction as follows:
“....... It is known that yields and biomass growth in grapevines are closely linked to their leaf area index (LAI, the total one-sided leaf area per unit of crop ground surface). ......”
Point 4: The sentence “The mean annual precipitation in this region is 16.5mm and the mean annual evaporation is 3600mm” is repeated in the introduction section, please remove one of them.
Response 4: Thank you for your suggestion. I have deleted it.
Point 5: Please add the space between vines within a row.
Response 5: Thank you for your suggestion. I described the average spacing between two vines in “2.1 Experimental fields” as follows:
“The average spacing between adjacent rows of vines was 1.3m, and the cultivation pattern and the schematic drawing for grape vines are shown in Figure1 and Figure 2.”
Point 6: Figure 1. Add growth stage according to BBCH scale at the moment of the picture or at least the date.
Response 6: Thank you for your suggestion. I added the growth stage according to BBCH scale as follows:
“Figure 1. Cultivation pattern of the grape vines in field. The principal growth stage is shoot development.”
Point 7: “In each case, the masses of the grapevine on the upper, lower, and middle shoots were recorded separately.” What do you mean with upper, lower, and middle shoots? Do you differentiate them for LAI estimation? If the calculation has been made with the average of the three specific locations, it should be made clear in the text. The trellis system used is not a conventional one, so a schematic of shoot locations can be useful for readers.
Response 7: Thank you for your suggestion. I added the schematic of shoot locations in Figure 1.
Point 8: “The method used to calculate LAI was defined by the function (1)”: Where did this method come from? Please add references. Also, you have measured “the main vein length of recorded leaves”. How do you transform this parameter to the leaf area used in Eq. 1?
Response 8: Thank you for your suggestion. Equation (1) is not correct, and I have revised it. I added Figure 3 to describe the relationship between the main vein length and the leaf area. The leaf area was measured by scanner. There is a power function relation between them.
Point 9: In the description of Equation 6, it says “k is the correction coefficient”, but there is no k in the formula.
Response 9: Thank you for your suggestion. I deleted it.
Point 10: In the equations, they name the experience coefficients as a, b and c. Sometimes different a, sometimes just b. Please be consistent in the way you express the equations. If they are not the same coefficient you can use different Greek letters, for example.
Response 10: Thank you for your suggestion. I have revised it.
Point 11: Here comes my major concern: Has the LAI estimation method (Eq 1) been validated in the study area? According to my experience and the bibliography consulted, the LAI on vineyards is up to 5 in very high dense vineyards. Thus, your results (LAIs up to 20) in an extremely arid region don’t seem realistic. Please review the formula used, or justified these results. You might need to consult more bibliography to justify your results. This applies to all the results from here on then measured LAI is used.
Response 11: Thank you for your suggestion. I have revised it.
Point 12: In Figure 3, please add to the footnote what LAI and CGDD/C mean. Figures and tables should be understandable without looking at the text. This comment applies to all Figures.
Response 12: Thank you for your suggestion. I revised it as follows:
“Figure 4. Relationship between leaf area index (LAI) and growing degree days (GDD) for different irrigation treatments.”
I also added the full name of abbreviation in all captions of Figures and Tables.
Point 13: In Table 3: I suppose Re/% is the relative or simulated error, please clarify in the table text. Maybe it can be used: E (%). (Also in tables 4 and 5).
Response 13: Thank you for your suggestion. Re is the relative error which defined in “2.5 Statistical analysis”. I added the explanation of abbreviation in the table text.
Point 14: In this section, it is stated: “Some of these factors are difficult to measure regularly, so the ability to compute the LAI using data that is readily obtained plays a very important role in making a given method useful in day-to-day work”. Can you provide methods, tools, or recommendations, at least based on the bibliography, to estimate the maximum LAI in a practical way?
Response 14: Thank you for your suggestion. The purpose of relationship between water consumption and LAIm is provide a method to estimate the LAIm. I added a method as an example as follows:
“. For example, LAIm can be measured directly when GDD is about 1690℃ as shown in Table 2. We selected the water consumption to simply estimate the LAIm. ”
Point 15: In Figure 7 footnote says: “Relationship between LAI and relative dry biomass for treatment X2 in 2009”. I understand this is for treatments X1-X6.
Response 15: Thank you for your suggestion. I have revised it.
Point 16: At the end of this section, it is said: “While any of the five models described in Table 2 can be used to obtain a reasonably accurate estimate of LAIm”. Table 2 is referred to the relative leaf area index (RLAI) and no model is described. Moreover, LAIm is the highest value of calculated LAI through Eq. 1, thus it is not estimated by the models presented.
Response 16: Thank you for your suggestion. LAIm should be revised LAI. I have revised it in the text.
Point 17: Highlight the minimum water consumption for the studied area.
Response 16: Thank you for your suggestion. I added the content as follows:
“(2) ...... In order to ensure the yields of grapevine during the growth period, the water consumption must be at least 132.46mm in the Turpan area.”
Reviewer 2 Report
The article as a whole attracts the reader's attention with its title. However, the title of the contribution does not completely correspond to its content (cultivation does not only include irrigation of the stand). The aim of the paper is not clearly and concisely defined. If the measurements were carried out in 2009, why they are published only now. The time lag may be related to the timeliness of the results. The methodological part of the thesis does not present and describe the method of determining LAI, which is a key parameter for the whole paper. It was a determination of LAI non-destructive or destructive method, with what error were the measurements performed? Habitat conditions (especially soil and climatic) should be described in detail. The work includes a sufficient number of variants related to irrigation, but measurements were performed for each variant in only 3 vines. If so, due to the high variability of the stand, this figure may not be entirely objective. I therefore recommend commenting on the methodology in detail. The authors use a number of abbreviations, which are specified in various parts of the text, which complicates the reader's orientation. The graphs used in the text must have their axes (X and Y) described consistently. Does the designation total biomass in graph 8 express the production of total biomass or only the yield of grapes? In the review of the literature, the authors often refer to old sources, this section would also deserve more attention. The conclusion of the paper should be better formulated, the main numerical values ​​should be stated and emphasized.
Author Response
Response to Reviewer 2 Comments
Point 1: If the measurements were carried out in 2009, why they are published only now. The time lag may be related to the timeliness of the results.
Response 1: Thank you for your suggestion. The data in manuscript were measured when I was doctoral student. Because of the lack of understanding about growth model of fruit tree, I did not use these data after graduating. But my research team have done more and more works about the crop (potato, winter wheat, summer maize, rice, and cotton) growth models in recent years, so we began to develop the growth model of fruit tree (Thompson Seedless grapevines, red globe grape and Zizyphus jujuba) again. I hope the time lag does not affect the significance of growth model for fruit tree.
Point 2: The methodological part of the thesis does not present and describe the method of determining LAI, which is a key parameter for the whole paper. It was a determination of LAI non-destructive or destructive method, with what error were the measurements performed? Habitat conditions (especially soil and climatic) should be described in detail.
Response 2: Thank you for your suggestion. We used the rulers to measure the length of shoot and the main vein length of leaves. It is a non-destructive method. LAI was calculated by Equation (1) as follows
,
And the leaf area (A) was calculated by the main vein length of leaf.
I added the measuring and calculating methods in “2.2. Experimental design”.
The soil condition was added as follows:
“The soil texture of experimental field is clay loam and uniform within one meter depth. The average soil bulk density is 1.47 g/cm3, and the average saturated soil water content is 0.39 cm3/cm3.”
Point 3: The work includes a sufficient number of variants related to irrigation, but measurements were performed for each variant in only 3 vines. If so, due to the high variability of the stand, this figure may not be entirely objective. I therefore recommend commenting on the methodology in detail.
Response 3: Thank you for your suggestion. I selected three grapevines in each treatment and three shoots in each selected grapevine as the fixed points to non-destructively observe. So there are 9 shoots in each treatment chosen to record. Finally, 9 groups of recorded data were averaged.
I added the detail of methodology in “2.2. Experimental design” and the schematic of selected shoot locations in Figure 1.
Point 4: The authors use a number of abbreviations, which are specified in various parts of the text, which complicates the reader's orientation.
Response 4: Thank you for your suggestion. In order to simply read, I added the full name of abbreviation in all captions of Figures and Tables.
Point 5: Does the designation total biomass in graph 8 express the production of total biomass or only the yield of grapes?
Response 4: Thank you for your suggestion. The total biomass includes the dry mass and the yield. I added the explanation in the caption of Figure 9.
Point 6: In the review of the literature, the authors often refer to old sources, this section would also deserve more attention.
Response 6: Thank you for your suggestion. I have revised the introduction and replaced the older references by the new references in recent ten years.
Point 7: The conclusion of the paper should be better formulated, the main numerical values should be stated and emphasized.
Response 7: Thank you for your suggestion. I highlight the minimum water consumption in conclusion as follows:
“(2) ...... In order to ensure the yields of grapevine during the growth period, the water consumption must be at least 132.46mm in the Turpan area.”
Round 2
Reviewer 1 Report
First of all, congratulations to the authors for greatly improving the manuscript in such a short time.
I have a few comments on some points that still should be addressed, and that is why I consider a minor revision:
Point 5. The authors have added the space between adjacent rows (1.2m), but the space between vines within a row (vines in the same row) is still missing.
Point 8. Thank you for correcting the error and recalculating the sections of the manuscript. However, in the reference used [47] the LAI is not calculated using equation 1. Please remove the reference or find a more suitable one.
In figure 6 the corresponding figure is crossed out, please check.
Check the reference list, it goes from 45 to 47 with a ??? in the middle.
Reviewer 2 Report
After studying the changes made to the text, I can state that most of the fundamental ambiguities have been supplemented and removed. Nevertheless, it would be appropriate to make a linguistic correction again.
